# Imagen Video: High Definition Video Generation with Diffusion Models

## Abstract

We present Imagen Video, a text-conditional video generation system based on a cascade of video diffusion models. Given a text prompt, Imagen Video generates high definition videos using a base video generation model and a sequence of interleaved spatial and temporal video super-resolution models. We describe how we scale up the system as a high definition text-to-video model including design decisions such as the choice of fully-convolutional temporal and spatial super-resolution models at certain resolutions, and the choice of the v-parameterization of diffusion models. In addition, we confirm and transfer findings from previous work on diffusion-based image generation to the video generation setting. Finally, we apply progressive distillation to our video models with classifier-free guidance for fast, high quality sampling. We find Imagen Video not only capable of generating videos of high fidelity, but also having a high degree of controllability and world knowledge, including the ability to generate diverse videos and text animations in various artistic styles and with 3D object understanding.

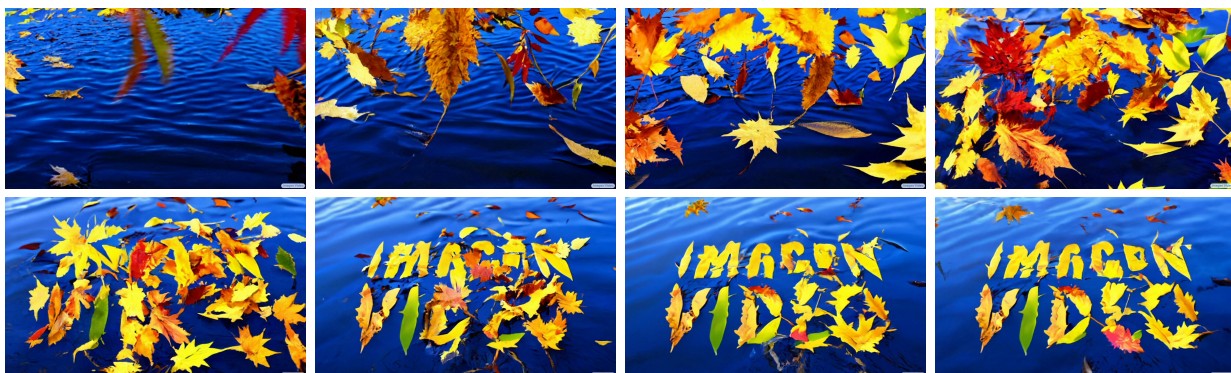

Figure 1: Imagen Video sample for the prompt: "*A bunch of autumn leaves falling on a calm lake to form the text 'Imagen Video'. Smooth.*" The generated video is at 1280×768 resolution, 5.3 second duration and 24 frames per second.

## 1 Introduction

Generative modeling has made tremendous progress with recent text-to-image systems like DALL-E 2 (Ramesh et al., 2022), Imagen (Saharia et al., 2022b), Parti (Yu et al., 2022), CogView (Ding et al., 2021) and Latent Diffusion (Rombach et al., 2022). Diffusion models (Sohl-Dickstein et al., 2015; Ho et al., 2020) in particular have found considerable success in multiple generative modeling tasks (Nichol & Dhariwal, 2021; Ho et al., 2022a; Dhariwal & Nichol, 2022) including density estimation (Kingma et al., 2021), text-to-speech (Chen et al., 2021a; Kong et al., 2021; Chen et al., 2021b), image-to-image (Saharia et al., 2022c;a; Whang et al., 2022), text-to-image (Rombach et al., 2022; Nichol et al., 2021; Ramesh et al., 2022; Saharia et al., 2022b) and 3D synthesis (Poole et al., 2022; Watson et al., 2022).

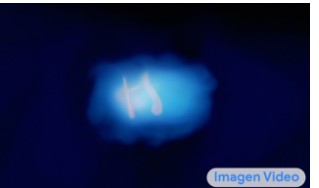 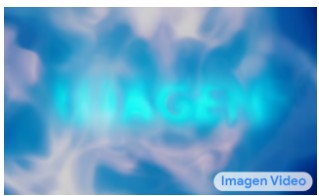 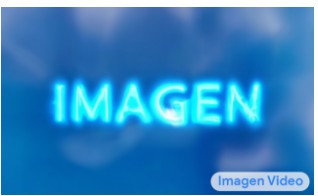 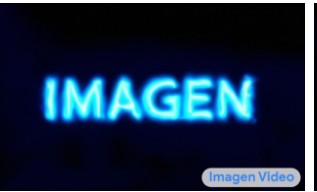 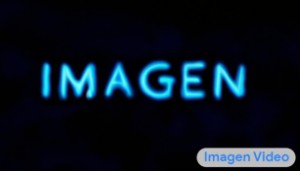

A colorful professional animated logo for 'Imagen Video' written using paint brush in cursive. Smooth animation.

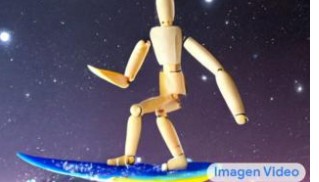 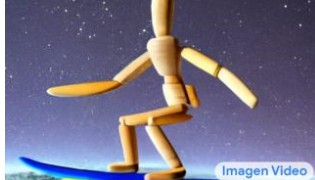 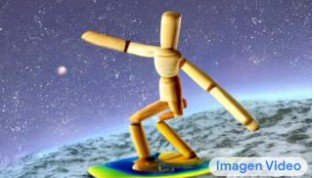 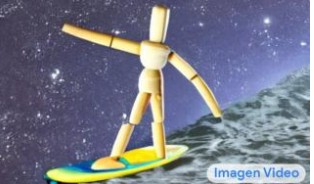 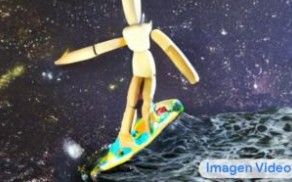

Blue flame transforming into the text "Imagen". Smooth animation

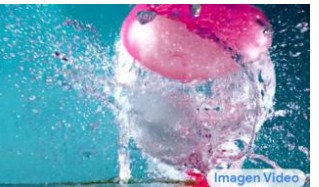 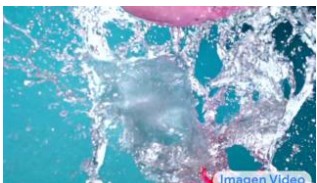 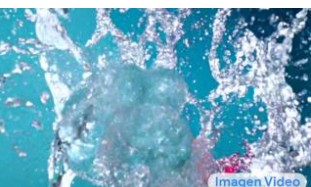 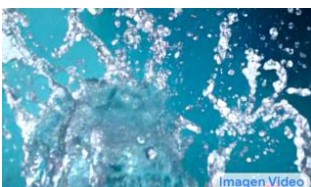 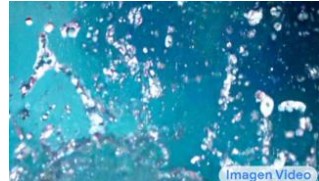

Wooden figurine surfing on a surfboard in space.

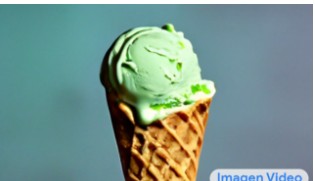 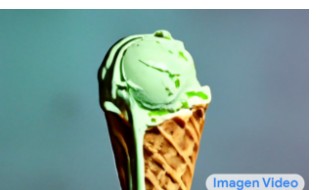 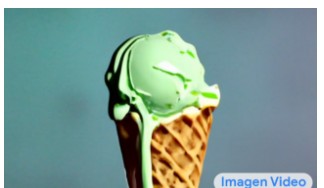 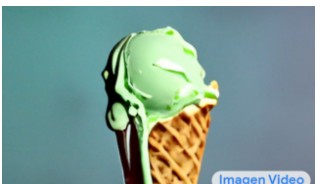 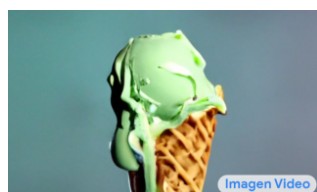

Balloon full of water exploding in extreme slow motion.

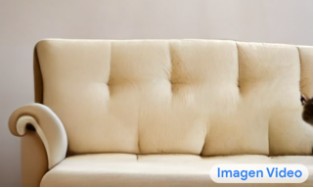 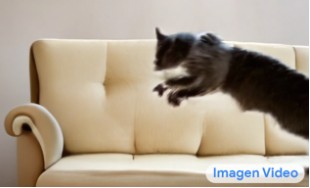 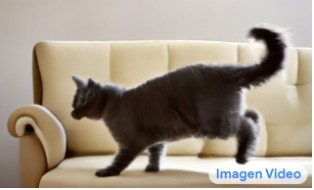 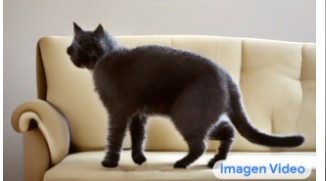 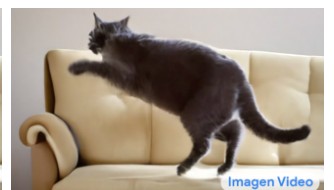

Melting pistachio ice cream dripping down the cone.

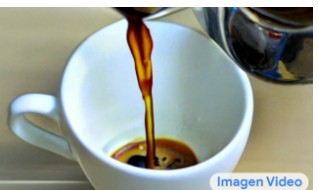 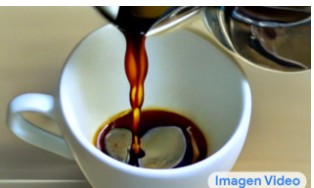 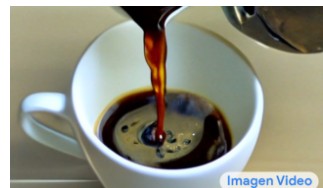 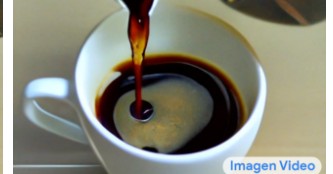 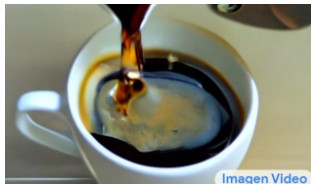

A british shorthair jumping over a couch.

Coffee pouring into a cup.

Figure 2: Videos generated from various text prompts. Imagen Video produces diverse and temporally-coherent videos that are well-aligned with the given prompt.

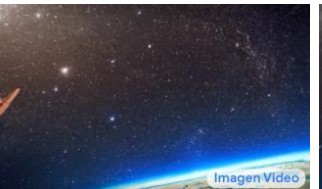 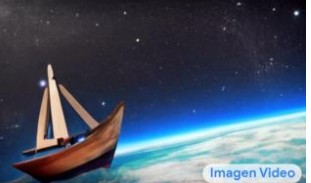 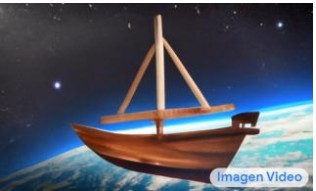 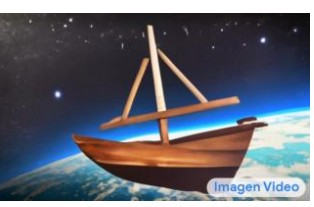 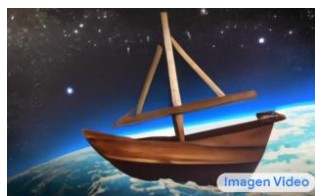

A small hand-crafted wooden boat taking off to space.

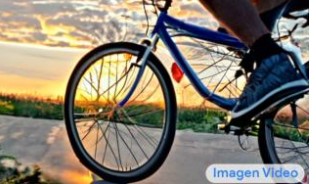 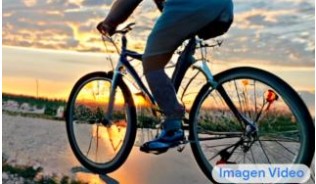 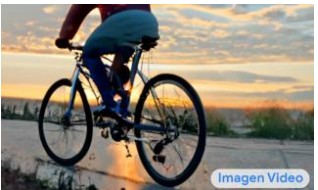 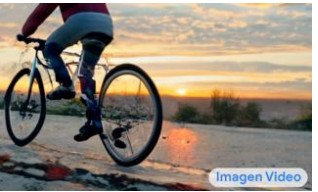 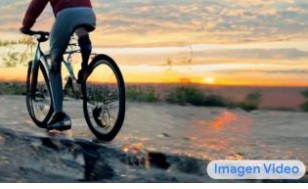

A person riding a bike in the sunset.

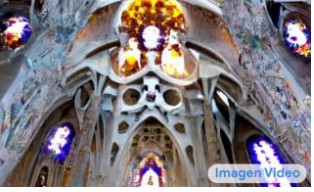 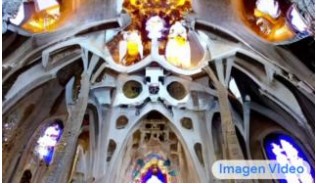 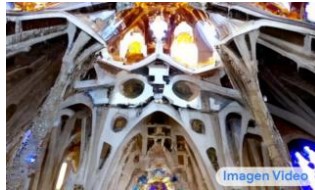 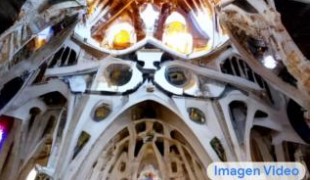 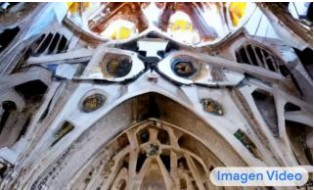

Drone flythrough interior of Sagrada Familia cathedral

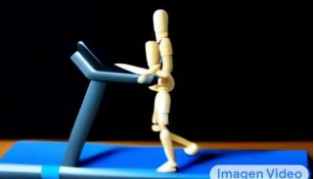 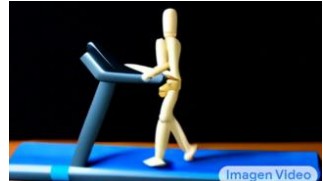 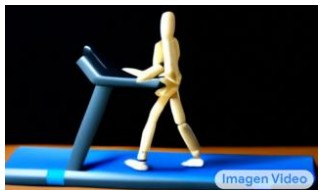 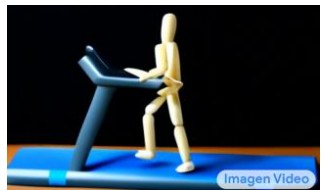 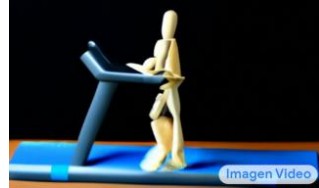

Wooden figurine walking on a treadmill made out of exercise mat.

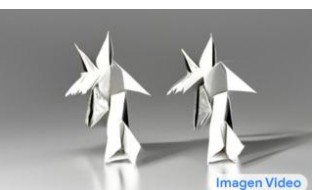 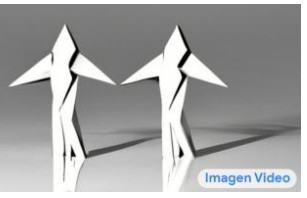 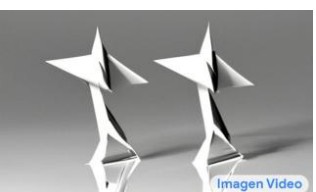 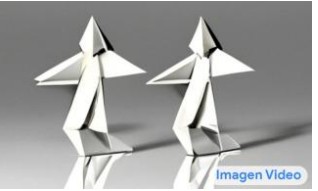 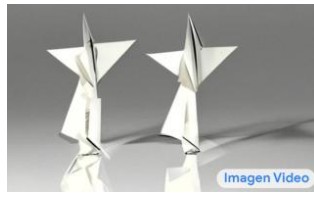

Origami dancers in white paper, 3D render, ultra-detailed, on white background, studio shot, dancing modern dance.

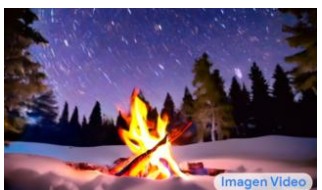 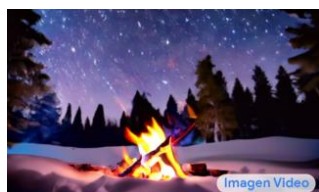 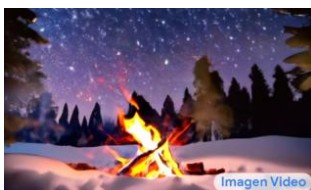 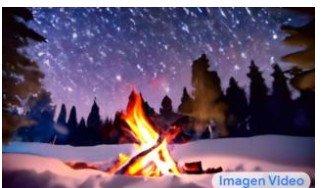 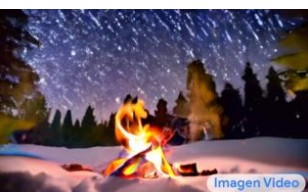

Campfire at night in a snowy forest with starry sky in the background.

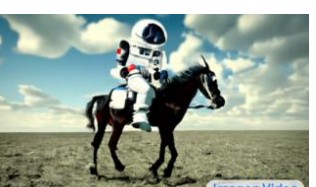 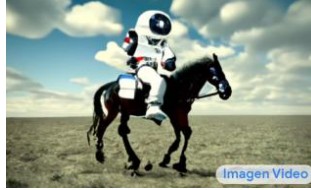 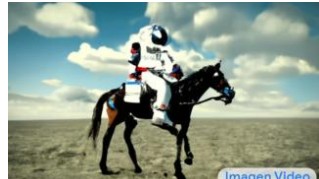 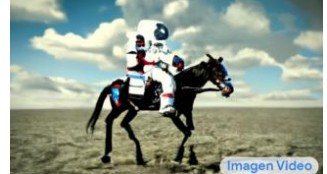 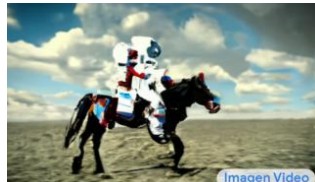

An astronaut riding a horse.

Figure 3: Videos generated from various text prompts. Imagen Video produces diverse and temporally-coherent videos that are well-aligned with the given prompt.

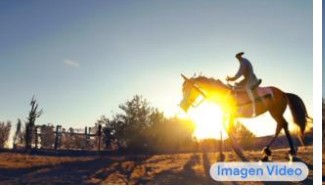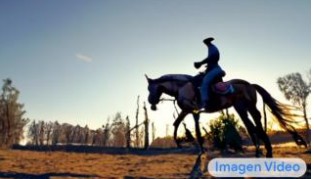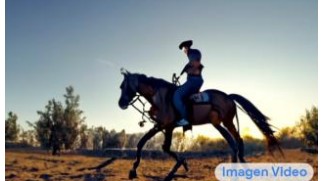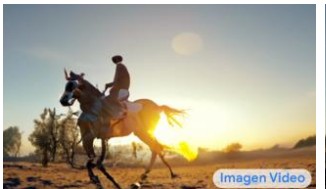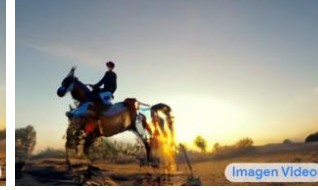

A person riding a horse in the sunrise.

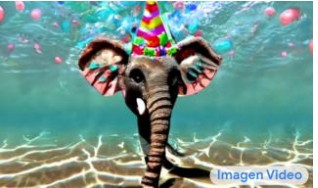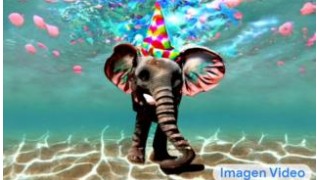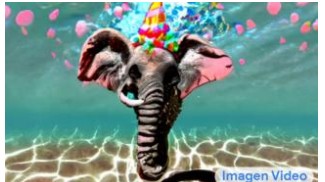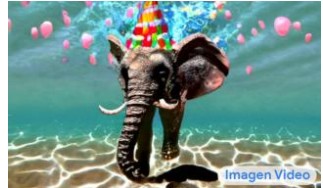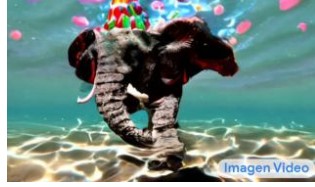

A happy elephant wearing a birthday hat walking under the sea.

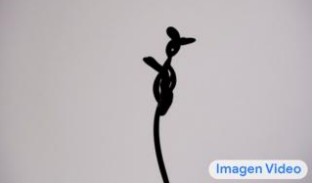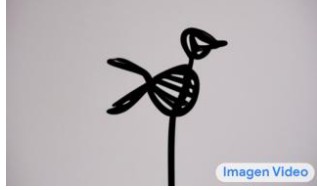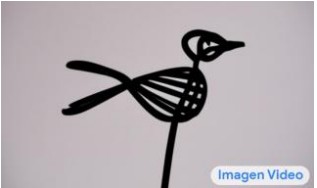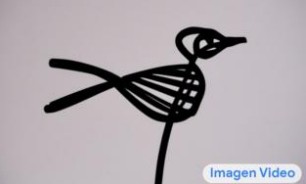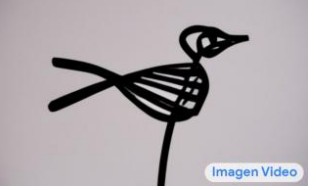

Studio shot of minimal kinetic sculpture made from thin wire shaped like a bird on white background.

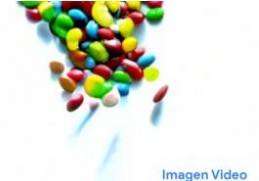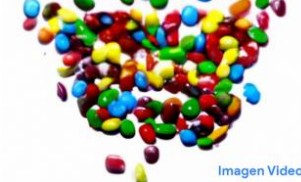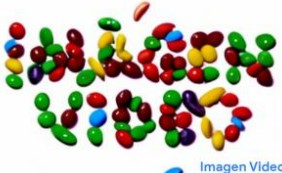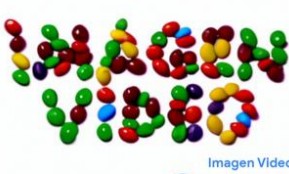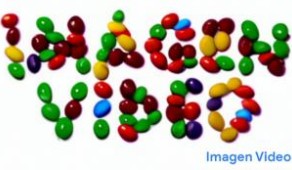

A bunch of colorful candies falling into a tray in the shape of text 'Imagen Video'. Smooth video.

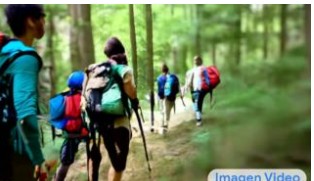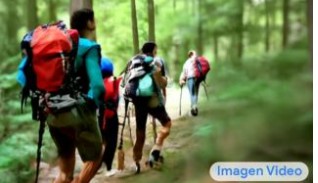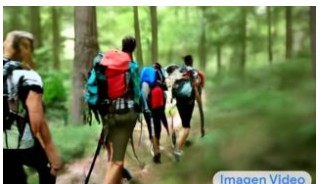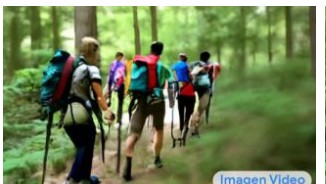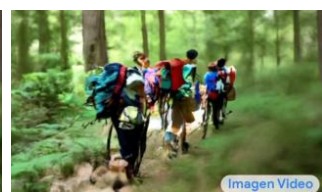

A group of people hiking in a forest.

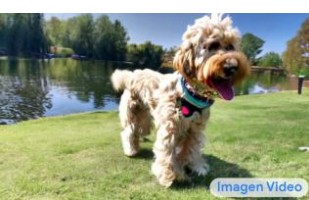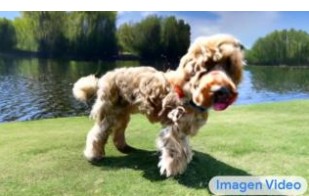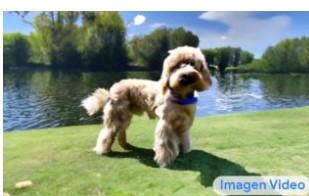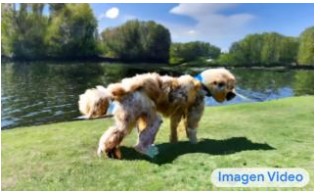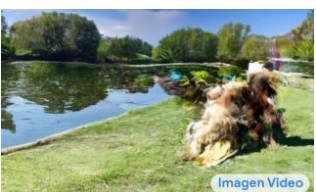

A goldendoodle playing in a park by a lake.

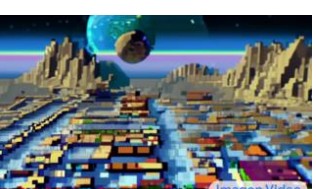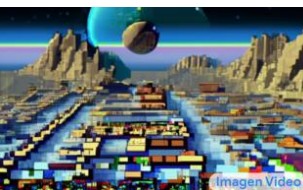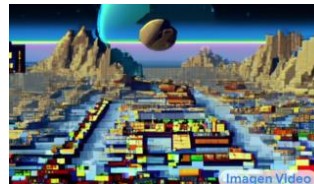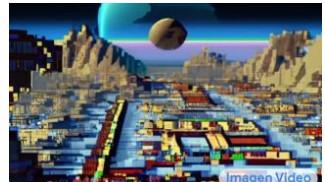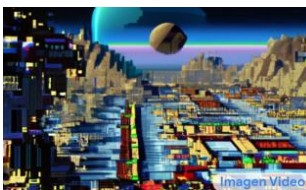

Incredibly detailed science fiction scene set on an alien planet, view of a marketplace. Pixel art.

Figure 4: Videos generated from various text prompts. Imagen Video produces diverse and temporally-coherent videos that are well-aligned with the given prompt.

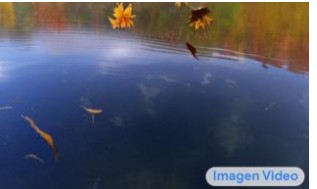 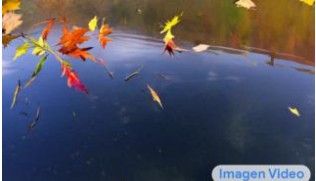 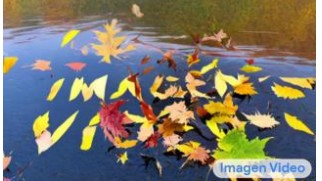 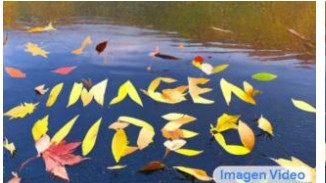 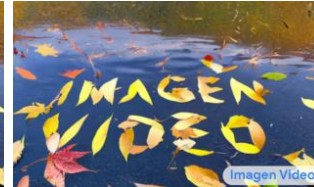

A bunch of autumn leaves falling on a calm lake to form the text 'Imagen Video'. Smooth.

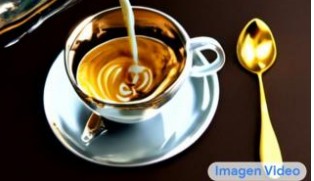 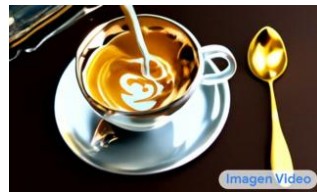 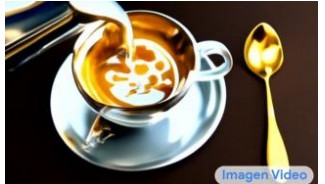 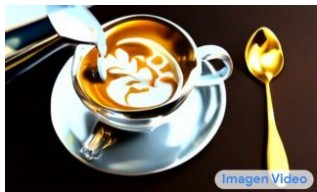 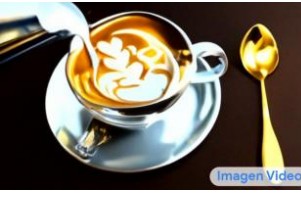

Pouring latte art into a silver cup with a golden spoon next to it.

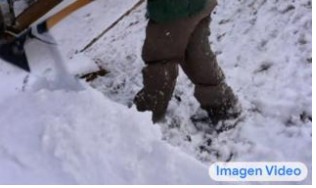 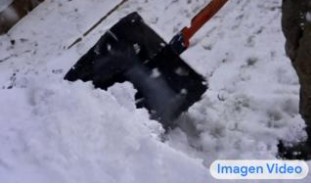 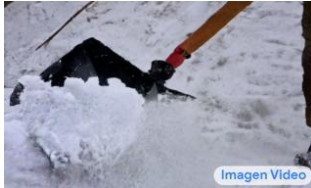 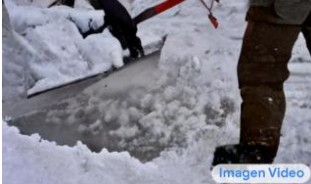 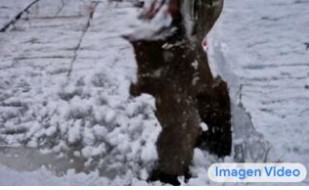

Shoveling snow.

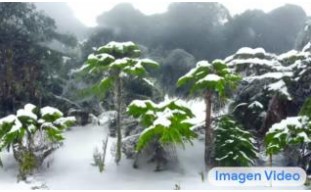 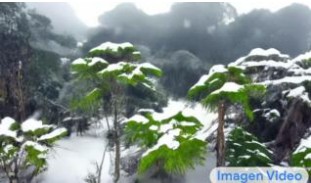 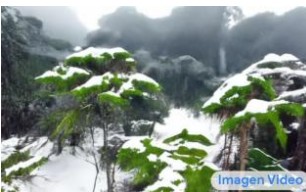 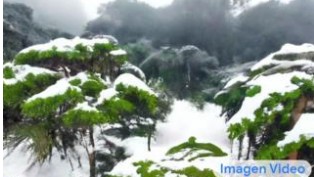 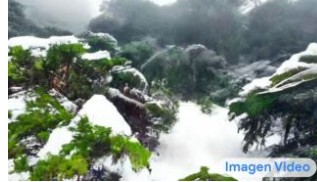

Drone flythrough of a tropical jungle covered in snow

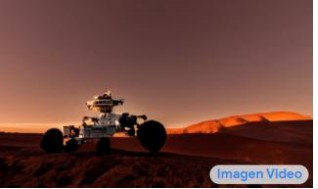 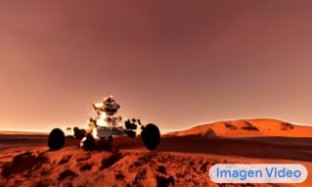 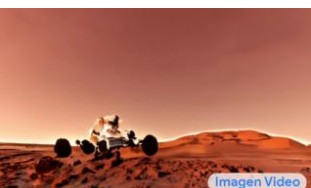 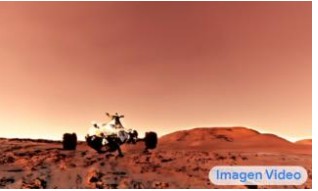 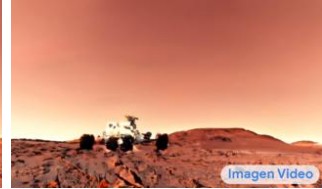

A beautiful sunrise on mars, Curiosity rover. High definition, timelapse, dramatic colors

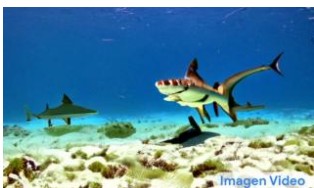 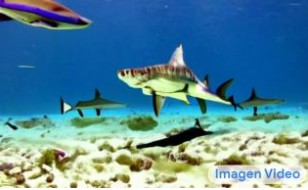 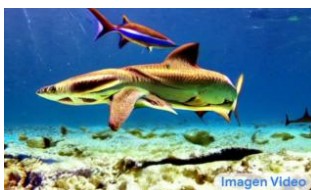 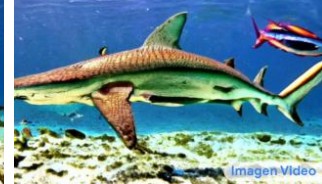 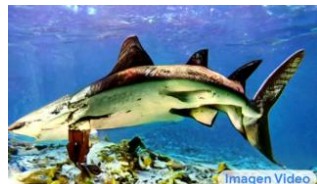

A shark swimming in clear Carribean ocean.

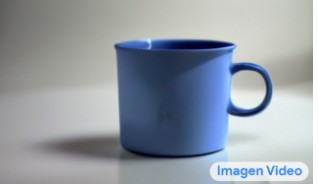 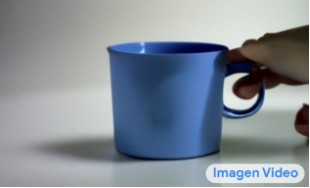 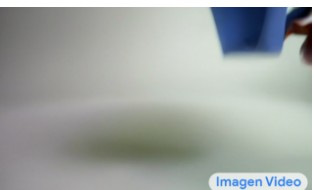 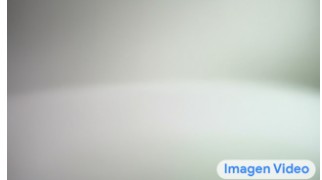 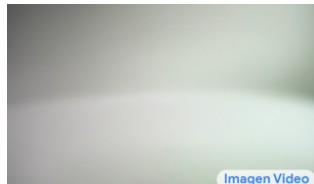

A hand lifts a cup.

Figure 5: Videos generated from various text prompts. Imagen Video produces diverse and temporally-coherent videos that are well-aligned with the given prompt.

Our work aims to generate videos from text. Prior work on video generation has focused on more restricted datasets with autoregressive models (Ranzato et al., 2014; Shi et al., 2015; Finn et al., 2016; Kalchbrenner et al., 2017; Babaeizadeh et al., 2021), latent-variable models with autoregressive priors (Mathieu et al., 2016; Vondrick et al., 2016; Babaeizadeh et al., 2018; Kumar et al., 2020), and more recently non-autoregressive latent-variable approaches (Gupta et al., 2022). Diffusion models have also shown promise for video generation (Ho et al., 2022b) at moderate resolution. Yang et al. (2022) showed autoregressive generation with a RNN-based model with conditional diffusion observations. The concurrent work of Singer et al. (2022) also applied text-to-video modelling with diffusion models, but built on a pretrained text-to-image model. Harvey et al. (2022) generates videos up to 25 minutes in length with video diffusion models, however the domain is restricted.

In this work, we introduce Imagen Video, a text-to-video generation system based on video diffusion models (Ho et al., 2022b) that is capable of generating high definition videos with high frame fidelity, strong temporal consistency, and deep language understanding. Imagen Video scales from prior work of 64-frame 128×128 videos at 24 frames per second to 128 frame 1280×768 high-definition video at 24 frames per second. Imagen Video has a simple architecture: The model consists of a frozen T5 text encoder (Raffel et al., 2020), a base video diffusion model, and interleaved spatial and temporal super-resolution diffusion models. Our key contributions are as follows:

1. We demonstrate the simplicity and effectiveness of cascaded diffusion video models for high definition video generation.

2. We confirm that recent findings in the text-to-image setting transfer to video generation, such as the effectiveness of frozen encoder text conditioning and classifier-free guidance.

3. We show new findings for video diffusion models that have implications for diffusion models in general, such as the effectiveness of the **v**-prediction parameterization for sample quality and the effectiveness of progressive distillation of guided diffusion models for the text-conditioned video generation setting.

4. We demonstrate qualitative controllability in Imagen Video, such as 3D object understanding, generation of text animations, and generation of videos in various artistic styles.

## 2 Imagen Video

Our model, *Imagen Video*, is a cascade of video diffusion models (Ho et al., 2022a;b). It consists of 7 sub-models which perform text-conditional video generation, spatial super-resolution, and temporal super-resolution. With the entire cascade, Imagen Video generates high definition 1280×768 (width × height) videos at 24 frames per second, for 128 frames ($\approx$ 5.3 seconds)—approximately 126 million pixels. We describe the components and techniques that constitute Imagen Video in the following sections.

### 2.1 Diffusion models

Imagen Video is built from diffusion models (Sohl-Dickstein et al., 2015; Song & Ermon, 2019; Ho et al., 2020) specified in continuous time (Tzen & Raginsky, 2019; Song et al., 2021; Kingma et al., 2021). We use the formulation of Kingma et al. (2021): the model is a latent variable model with latents $\mathbf{z} = \{\mathbf{z}_t \mid t \in [0,1]\}$ following a forward process $q(\mathbf{z}|\mathbf{x})$ starting at data $\mathbf{x} \sim p(\mathbf{x})$. The forward process is a Gaussian process that satisfies the Markovian structure:

$$q(\mathbf{z}_t|\mathbf{x}) = \mathcal{N}(\mathbf{z}_t; \alpha_t \mathbf{x}, \sigma_t^2 \mathbf{I}), \quad q(\mathbf{z}_t|\mathbf{z}_s) = \mathcal{N}(\mathbf{z}_t; (\alpha_t/\alpha_s)\mathbf{z}_s, \sigma_{t|s}^2 \mathbf{I}) \tag{1}$$

where $0 \leq s < t \leq 1$, $\sigma_{t|s}^2 = (1 - e^{\lambda_t - \lambda_s})\sigma_t^2$, and $\alpha_t, \sigma_t$ specify a noise schedule whose log signal-to-noise-ratio $\lambda_t = \log[\alpha_t^2/\sigma_t^2]$ decreases monotonically with $t$ until $q(\mathbf{z}_1) \approx \mathcal{N}(\mathbf{0}, \mathbf{I})$. We use a continuous time version of the cosine noise schedule (Nichol & Dhariwal, 2021). The generative model is a learned model that matches this forward process in the reverse time direction, generating $\mathbf{z}_t$ starting from $t = 1$ and ending at $t = 0$.

Learning to reverse the forward process for generation can be reduced to learning to denoise $\mathbf{z}_t \sim q(\mathbf{z}_t|\mathbf{x})$ into an estimate $\hat{\mathbf{x}}_\theta(\mathbf{z}_t, \lambda_t) \approx \mathbf{x}$ for all $t$. Like (Song & Ermon, 2019; Ho et al., 2020) and most follow-up work, we optimize the model by minimizing a simple noise-prediction loss:

$$\mathcal{L}(\mathbf{x}) = \mathbb{E}_{\boldsymbol{\epsilon} \sim \mathcal{N}(0,\mathbf{I}), t \sim U(0,1)} \left[ \|\hat{\boldsymbol{\epsilon}}_\theta(\mathbf{z}_t, \lambda_t) - \boldsymbol{\epsilon}\|_2^2 \right] \tag{2}$$

where $\mathbf{z}_t = \alpha_t \mathbf{x} + \sigma_t \boldsymbol{\epsilon}$, and $\hat{\boldsymbol{\epsilon}}_\theta(\mathbf{z}_t, \lambda_t) = \sigma_t^{-1}(\mathbf{z}_t - \alpha_t \hat{\mathbf{x}}_\theta(\mathbf{z}_t, \lambda_t))$. We will drop the dependence on $\lambda_t$ to simplify notation. In practice, we parameterize our models in terms of the $\mathbf{v}$-parameterization (Salimans & Ho, 2022), rather than predicting $\boldsymbol{\epsilon}$ or $\mathbf{x}$ directly; see Section 2.4.

For conditional generative modeling, we provide the conditioning information $\mathbf{c}$ drawn jointly with $\mathbf{x}$ to the model as $\hat{\mathbf{x}}_\theta(\mathbf{z}_t, \mathbf{c}_t)$. We use these conditional diffusion models for spatial and temporal super-resolution in our pipeline of diffusion models: in these cases, $\mathbf{c}$ includes both the text and the previous stage low resolution video as well as a signal $\lambda_t'$ that describes the strength of conditioning augmentation added to $\mathbf{c}$. Saharia et al. (2022b) found it critical to condition all the super-resolution models with the text embedding, and we follow this approach.

We use the discrete time ancestral sampler (Ho et al., 2020), with sampling variances derived from lower and upper bounds on reverse process entropy (Sohl-Dickstein et al., 2015; Ho et al., 2020; Nichol & Dhariwal, 2021). This sampler can be formulated by using a reversed description of the forward process as $q(\mathbf{z}_s|\mathbf{z}_t, \mathbf{x}) = \mathcal{N}(\mathbf{z}_s; \tilde{\boldsymbol{\mu}}_{s|t}(\mathbf{z}_t, \mathbf{x}), \tilde{\sigma}_{s|t}^2 \mathbf{I})$ (noting $s < t$), where

$$\tilde{\boldsymbol{\mu}}_{s|t}(\mathbf{z}_t, \mathbf{x}) = e^{\lambda_t - \lambda_s}(\alpha_s/\alpha_t)\mathbf{z}_t + (1 - e^{\lambda_t - \lambda_s})\alpha_s \mathbf{x} \quad \text{and} \quad \tilde{\sigma}_{s|t}^2 = (1 - e^{\lambda_t - \lambda_s})\sigma_s^2. \tag{3}$$

Starting at $\mathbf{z}_1 \sim \mathcal{N}(\mathbf{0}, \mathbf{I})$, the ancestral sampler follows the rule

$$\mathbf{z}_s = \tilde{\boldsymbol{\mu}}_{s|t}(\mathbf{z}_t, \hat{\mathbf{x}}_\theta(\mathbf{z}_t)) + \sqrt{(\tilde{\sigma}_{s|t}^2)^{1-\gamma}(\sigma_{t|s}^2)^\gamma}\, \boldsymbol{\epsilon} \tag{4}$$

where $\boldsymbol{\epsilon}$ is standard Gaussian noise, $\gamma$ is a hyperparameter that controls the stochasticity of the sampler (Nichol & Dhariwal, 2021), and $s, t$ follow a uniformly spaced sequence from 1 to 0. See Section 3 for sampler hyperparameter settings.

Alternatively, the deterministic DDIM sampler (Song et al., 2020) can be used for sampling. This sampler is a numerical integration rule for the probability flow ODE (Song et al., 2021; Salimans & Ho, 2022), which describes how a sample from a standard normal distribution can be deterministically transformed into a sample from the video data distribution using the denoising model. The DDIM sampler is useful for progressive distillation for fast sampling, as described in Section 2.7.

## 2.2 Cascaded Diffusion Models and text conditioning

Cascaded Diffusion Models (Ho et al., 2022a) are an effective method for scaling diffusion models to high resolution outputs, finding considerable success in both class-conditional ImageNet (Ho et al., 2022a) and text-to-image generation (Ramesh et al., 2022; Saharia et al., 2022b). Cascaded diffusion models generate an image or video at a low resolution, then sequentially increase the resolution of the image or video through a series of super-resolution diffusion models. Cascaded Diffusion Models can model very high dimensional problems while still keeping each sub-model relatively simple. *Imagen* (Saharia et al., 2022b) also showed that by conditioning on text embeddings from a large frozen language model in conjunction with cascaded diffusion models, one can generate high quality $1024 \times 1024$ images from text descriptions. In this work we extend this approach to video generation.

Figure 6 summarizes the entire cascading pipeline of Imagen Video. In total, we have 1 frozen text encoder, 1 base video diffusion model, 3 SSR (spatial super-resolution), and 3 TSR (temporal super-resolution) models – for a total of 7 video diffusion models, with a total of 11.6B diffusion model parameters. The data used to train these models is processed to the appropriate spatial and temporal resolutions by spatial resizing and frame skipping. At generation time, the SSR models increase spatial resolution for all input frames, whereas the TSR models increase temporal resolution by filling in intermediate frames between input frames. All

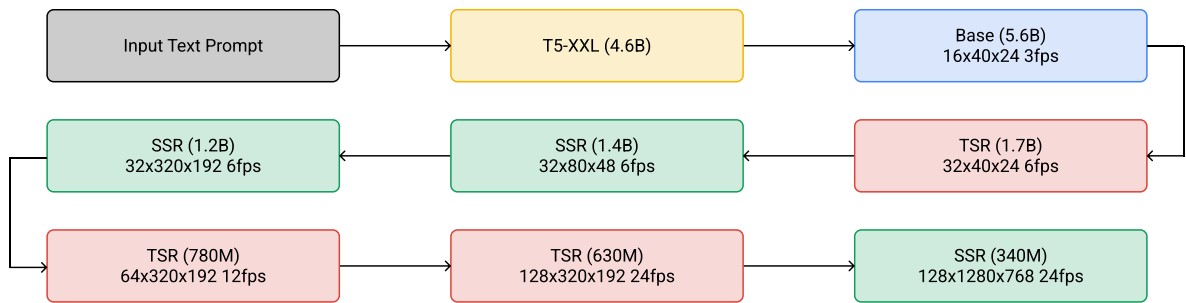

Figure 6: The cascaded sampling pipeline starting from a text prompt input to generating a 5.3-second, 1280×768 video at 24fps. "SSR" and "TSR" denote spatial and temporal super-resolution respectively, and videos are labeled as frames×width×height. In practice, the text embeddings are injected into all models, not just the base model.

models generate an entire block of frames simultaneously – so for instance, our SSR models do not suffer from obvious artifacts that would occur from naively running super-resolution on independent frames.

One benefit of cascaded models is that each diffusion model can be trained independently, allowing one to train all 7 models in parallel. Additionally, our super-resolution models are general purpose video super-resolution models, and they can be applied to real videos or samples from generative models other than the ones presented in this paper. This is similar to how Imagen's super-resolution models helped improve the fidelity of the images generated by Parti (Yu et al., 2022), which is an autoregressive text-to-image model. We intend to explore hybrid pipelines of multiple model classes further in future work.

Similar to Saharia et al. (2022b), we utilize contextual embeddings from a frozen T5-XXL text encoder (Raffel et al., 2020) for conditioning on the input text prompt. We find these embeddings to be critical for alignment between generated video and the text prompt. Similar to the findings of Saharia et al. (2022b), we observe evidence of deeper language understanding, enabling us to generate the videos displayed in Figs. 2 to 5.

## 2.3 Video diffusion architectures

Diffusion models for image generation typically use a 2D U-Net architecture (Ronneberger et al., 2015; Salimans et al., 2017; Ho et al., 2020) to represent the denoising model $\hat{\mathbf{x}}_\theta$. This is a multiscale model consisting of multiple layers of spatial attention and convolution at each resolution, combined with shortcuts between layers at the same resolution. In earlier work on *Video Diffusion Models*, Ho et al. (2022b) introduced the Video U-Net , which generalizes the 2D diffusion model architecture to 3D in a space-time separable fashion using temporal attention and convolution layers interleaved within spatial attention and convolution layers to capture dependencies between video frames. Our work builds on the Video U-Net architecture: see Figure 7. Following Video Diffusion Models, each of our denoising models $\hat{\mathbf{x}}_\theta$ operate on multiple video frames simultaneously and thereby generate entire blocks of video frames at a time, which we find to be important to capture the temporal coherence of the generated video compared to frame-autoregressive approaches. Our spatial super-resolution (SSR) and temporal super-resolution (TSR) models condition on their input videos by concatenating an upsampled conditioning input channelwise to the noisy data $\mathbf{z}_t$, the same mechanism as SR3 (Saharia et al., 2022c) and Palette (Saharia et al., 2022a): spatial upsampling before concatenation is performed using bilinear resizing, and temporal upsampling before concatenation is performed by repeating frames or by filling in blank frames.

Our base video model, which is the first model in the pipeline that generates data at the lowest frame count and spatial resolution, uses temporal attention to mix information across time. Our SSR and TSR models, on the other hand, use temporal convolutions instead of temporal attention. The temporal attention in the base

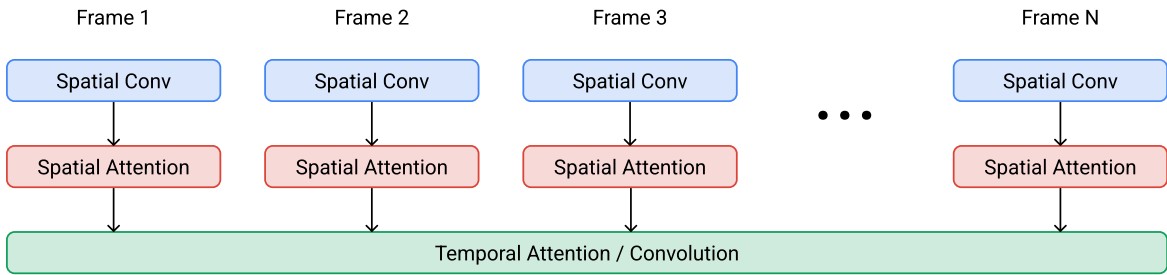

Figure 7: Video U-Net space-time separable block. Spatial operations are performed independently over frames with shared parameters, whereas the temporal operation mixes activations over frames. Our base model uses spatial convolutions, spatial self-attention and temporal self-attention. For memory efficiency, our spatial and temporal super-resolution models use temporal convolutions instead of attention, and our models at the highest spatial resolution do not have spatial attention.

model enables Imagen Video to model long term temporal dependencies, while the temporal convolutions in the SSR and TSR models allow Imagen Video to maintain local temporal consistency during upsampling. The use of temporal convolutions lowers memory and computation costs over temporal attention—this is crucial because the very purpose of the TSR and SSR models is to operate at high frame rates and spatial resolutions. In our initial experiments, we did not find any significant improvements when using temporal attention over temporal convolutions in our SSR and TSR models, which we hypothesize is due to the significant amount of temporal correlation already present in the conditioning input to these models.

Our models also use spatial attention and spatial convolutions. The base model and the first two spatial super-resolution models have spatial attention in addition to spatial convolutions. We found this to improve sample fidelity. However, as we move to higher resolutions, we switch to fully convolutional architectures, like Saharia et al. (2022b), to minimize memory and compute costs in order to generate $1280 \times 768$ resolution data. The highest resolution SSR model in our pipeline is a fully convolutional model trained on random lower resolution spatial crops for training time memory efficiency, and we find that the model easily generalizes to the full resolution during sampling time.

## 2.4 v-prediction

We follow Salimans & Ho (2022) and use $\mathbf{v}$-prediction parameterization ($\mathbf{v}_t \equiv \alpha_t \boldsymbol{\epsilon} - \sigma_t \mathbf{x}$) for all our models. The $\mathbf{v}$-parameterization is particularly useful for numerical stability throughout the diffusion process to enable progressive distillation for our models. For models that operate at higher resolution in our pipeline, we also discovered that the $\mathbf{v}$-parameterization avoids color shifting artifacts that are known to affect high resolution diffusion models, and in the video setting it avoids temporal color shifting that sometimes appears with $\boldsymbol{\epsilon}$-prediction models. Our use of $\mathbf{v}$-parameterization also has the benefit of faster convergence of sample quality metrics: see Section 3.3.

## 2.5 Conditioning Augmentation

We use noise conditioning augmentation (Ho et al., 2022a) for all our temporal and spatial super-resolution models. Noise conditioning augmentation has been found to be critical for cascaded diffusion models for class-conditional generation (Ho et al., 2022a) as well as text-to-image models (Saharia et al., 2022b). In particular, it facilitates parallel training of different models in the cascade, as it reduces the sensitivity to domain gaps between the output of one stage of the cascade and the inputs used in training the subsequent stage.

Following Ho et al. (2022a), we apply Gaussian noise augmentation with a random signal-to-noise ratio to the conditioning input video during training, and this sampled signal-to-noise ratio is provided to the model

as well. At sampling time we use a fixed signal-to-noise ratio such as 3 or 5, representing a small amount of augmentation that aids in removing artifacts in the samples from the previous stage while preserving most of the structure.

## 2.6 Video-Image Joint Training

We follow Ho et al. (2022b) in jointly training all the models in the Imagen Video pipeline on images and videos. During training, individual images are treated as single frame videos. We achieve this by packing individual independent images into a sequence of the same length as a video, and bypass the temporal convolution residual blocks by masking out their computation path. Similarly, we disable cross-frame temporal attention by applying masking to the temporal attention maps. This strategy allows us to use to train our video models on image-text datasets that are significantly larger and more diverse than available video-text datasets. Consistent with Ho et al. (2022b), we observe that joint training with images significantly increases the overall quality of video samples. Another interesting artifact of joint training is the knowledge transfer from images to videos. For instance, while training on natural video data only enables the model to learn dynamics in natural settings, the model can learn about different image styles (such as sketch, painting, etc.) by training on images. As a result, this joint training enables the model to generate interesting video dynamics in different styles. See Fig. 8 for such examples.

### 2.6.1 Classifier Free Guidance

We found classifier free guidance (Ho & Salimans, 2021) to be critical for generating high fidelity samples which respect a given text prompt. This is consistent with earlier results on text-to-image models (Nichol et al., 2021; Ramesh et al., 2022; Saharia et al., 2022b; Yu et al., 2022).

In the conditional generation setting, the data $\mathbf{x}$ is generated conditional on a signal $\mathbf{c}$, which here represents a contextualized embedding of the text prompt, and a conditional diffusion model can be trained by using this signal $\mathbf{c}$ as an additional input to the denoising model $\hat{\mathbf{x}}_\theta(\mathbf{z}_t, \mathbf{c})$. After training, Ho & Salimans (2021) find that sample quality can be improved by adjusting the denoising prediction $\hat{\mathbf{x}}_\theta(\mathbf{z}_t, \mathbf{c})$ using

$$\tilde{\mathbf{x}}_\theta(\mathbf{z}_t, \mathbf{c}) = (1 + w)\hat{\mathbf{x}}_\theta(\mathbf{z}_t, \mathbf{c}) - w\hat{\mathbf{x}}_\theta(\mathbf{z}_t), \tag{5}$$

where $w$ is the *guidance strength*, $\hat{\mathbf{x}}_\theta(\mathbf{z}_t, \mathbf{c})$ is the conditional model, and $\hat{\mathbf{x}}_\theta(\mathbf{z}_t) = \hat{\mathbf{x}}_\theta(\mathbf{z}_t, \mathbf{c} = \emptyset)$ is an unconditional model. The unconditional model is jointly trained with the conditional model by dropping out the conditioning input $\mathbf{c}$. The predictions of the adjusted denoising model $\tilde{\mathbf{x}}_\theta(\mathbf{z}_t, \mathbf{c})$ are clipped to respect the range of possible pixel values, which we discuss in more detail in the next section. Note that the linear transformation in Equation 5 can equivalently be performed in $\mathbf{v}$-space ($\tilde{\mathbf{v}}_\theta(\mathbf{z}_t, \mathbf{c}) = (1 + w)\hat{\mathbf{v}}_\theta(\mathbf{z}_t, \mathbf{c}) - w\hat{\mathbf{v}}_\theta(\mathbf{z}_t)$) or $\epsilon$-space ($\tilde{\epsilon}_\theta(\mathbf{z}_t, \mathbf{c}) = (1 + w)\hat{\epsilon}_\theta(\mathbf{z}_t, \mathbf{c}) - w\hat{\epsilon}_\theta(\mathbf{z}_t)$).

For $w > 0$ this adjustment has the effect of over-emphasizing the effect of conditioning on the signal $\mathbf{c}$, which tends to produce samples of lower diversity but higher quality compared to sampling from the regular conditional model (Ho & Salimans, 2021). The method can be interpreted as a way to guide the samples towards areas where an implicit classifier $p(\mathbf{c}|\mathbf{z}_t)$ has high likelihood; as such, it is an adaptation of the explicit classifier guidance method proposed by Dhariwal & Nichol (2022).

### 2.6.2 Large Guidance Weights

When using large guidance weights, the resulting $\tilde{\mathbf{x}}_\theta(\mathbf{z}_t, \mathbf{c})$ must be projected back to the possible range of pixel values at every sampling step to prevent train-test mismatch. When using large guidance weights, the standard approach, i.e., clipping the values to the right range (e.g., `np.clip(x, -1, 1)`), leads to significant saturation artifacts in the generated videos. A similar effect was observed in Saharia et al. (2022b) for text-to-image generation. Saharia et al. (2022b) use *dynamic thresholding* to alleviate this saturation issue. Specifically, dynamic clipping involves clipping the image to a dynamically chosen threshold `s` followed by scaling by `s` (i.e., `np.clip(x, -s, s) / s`) (Saharia et al., 2022b).

Although dynamic clipping can help with over-saturation, we did not find it sufficient in initial experiments. We therefore also experiment with letting $w$ oscillate between a high and a low guidance weight at each

alternating sampling step, which we find significantly helps with these saturation issues. We call this sampling technique *oscillating guidance.* Specifically, we use a constant high guidance weight for a certain number of initial sampling steps, followed by oscillation between high and low guidance weights: this oscillation is implemented simply by alternating between a large weight (such as 15) and a small weight (such as 1) over the course of sampling. We hypothesize that a constant high guidance weight at the start of sampling helps break modes with heavy emphasis on text, while oscillating between high and low guidance weights helps maintain a strong text alignment (via high guidance sampling step) while limiting saturation artifacts (via low guidance sampling step). We however observed no improvement in sample fidelity and more visual artifacts when applying oscillating guidance to models past the $80{\times}48$ spatial resolution. Thus we only apply oscillating guidance to the base and the first two SR models.

## 2.7 Progressive Distillation with Guidance and Stochastic Samplers

Salimans & Ho (2022) proposed *progressive distillation* to enable fast sampling of diffusion models. This method distills a trained deterministic DDIM sampler (Song et al., 2020) to a diffusion model that takes many fewer sampling steps, without losing much perceptual quality. At each iteration of the distillation process, an $N$-step DDIM sampler is distilled to a new model with $N/2$-steps. This procedure is repeated by halving the required sampling steps each iteration. Meng et al. (2022) extend this approach to samplers with guidance, and propose a new stochastic sampler for use with distilled models. Here we show that this approach also works very well for video generation.

We use a two-stage distillation approach to distill a DDIM sampler (Song et al., 2020) with classifier-free guidance. At the first stage, we learn a single diffusion model that matches the combined output from the jointly trained conditional and unconditional diffusion models, where the combination coefficients are determined by the guidance weight. Then we apply progressive distillation to that single model to produce models requiring fewer sampling steps at the second stage.

After distillation, we use a stochastic $N$-step sampler: At each step, we first apply one deterministic DDIM update with twice the original step size (i.e., the same step size as a $N/2$-step sampler), and then we perform one stochastic step backward (i.e., perturbed with noise following the forward diffusion process) with the original step size, inspired by Karras et al. (2022). See Meng et al. (2022) for more details. Using this approach, we are able to distill all 7 video diffusion models down to just 8 sampling steps per model without any noticeable loss in perceptual quality.

# 3 Experiments

We train our models on a combination of an internal dataset consisting of 14 million video-text pairs and 60 million image-text pairs, and the publicly available LAION-400M image-text dataset (Schuhmann et al., 2021). To process the data into a form suitable for training our cascading pipeline, we spatially resize images and videos using antialiased bilinear resizing, and we temporally resize videos by skipping frames. Throughout our model development process, we evaluated Imagen Video on several different metrics, such as FID on individual frames (Heusel et al., 2017), FVD (Unterthiner et al., 2019) for temporal consistency, and frame-wise CLIP scores (Hessel et al., 2021; Park et al., 2021) for video-text alignment. Below, we explore the capabilities of our model and investigate its performance in regards to 1) *scaling up* the number of parameters in our model, 2) changing the *parameterization* of our model, and 3) *distilling* our models so that they are fast to sample from.

## 3.1 Unique video generation capabilities

We find that Imagen Video is capable of generating high fidelity video, and that it possesses several unique capabilities that are not traditionally found in unstructured generative models learned purely from data. For example, Fig. 8 shows that our model is capable of generating videos with artistic styles learned from image information, such as videos in the style of van Gogh paintings or watercolor paintings. Fig. 9 shows that Imagen Video possesses an understanding of 3D structure, as it is capable of generating videos of objects rotating while roughly preserving structure. While the 3D consistency over the course of rotation is not

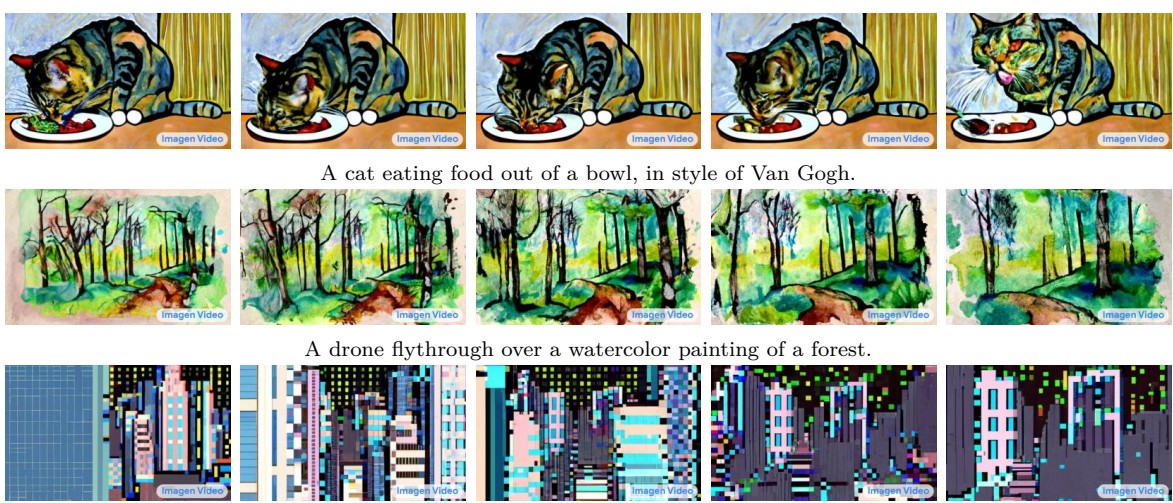

A cat eating food out of a bowl, in style of Van Gogh.

A drone flythrough over a watercolor painting of a forest.

Drone flythrough of a pixel art of futuristic city.

Figure 8: Snapshots of frames from videos generated by Imagen Video demonstrating the ability of the model to generate dynamics in different artistic styles.

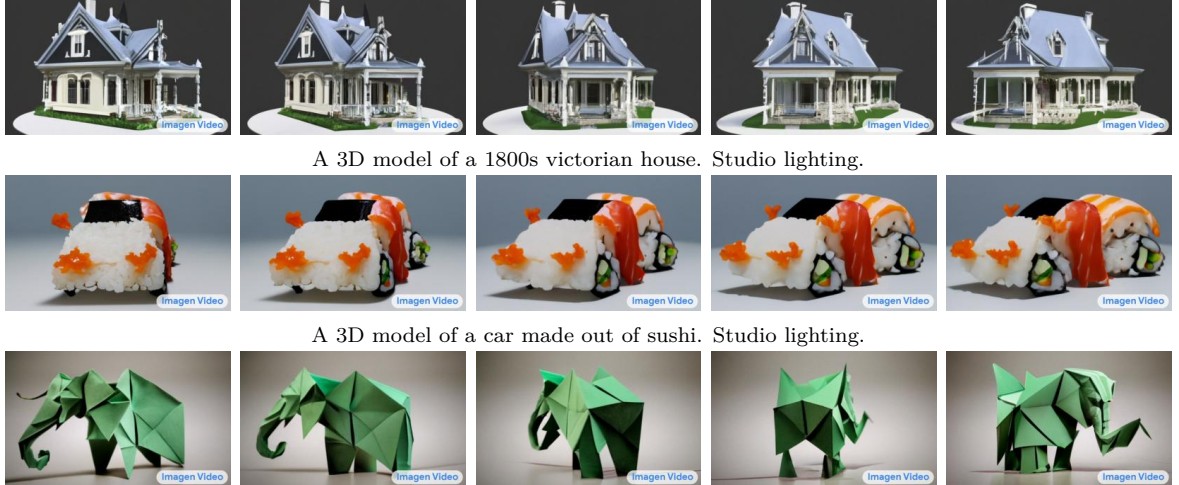

A 3D model of a 1800s victorian house. Studio lighting.

A 3D model of a car made out of sushi. Studio lighting.

A 3D model of an elephant origami. Studio lighting.

Figure 9: Snapshots of frames from videos generated by Imagen Video demonstrating the model's understanding of 3D structures.

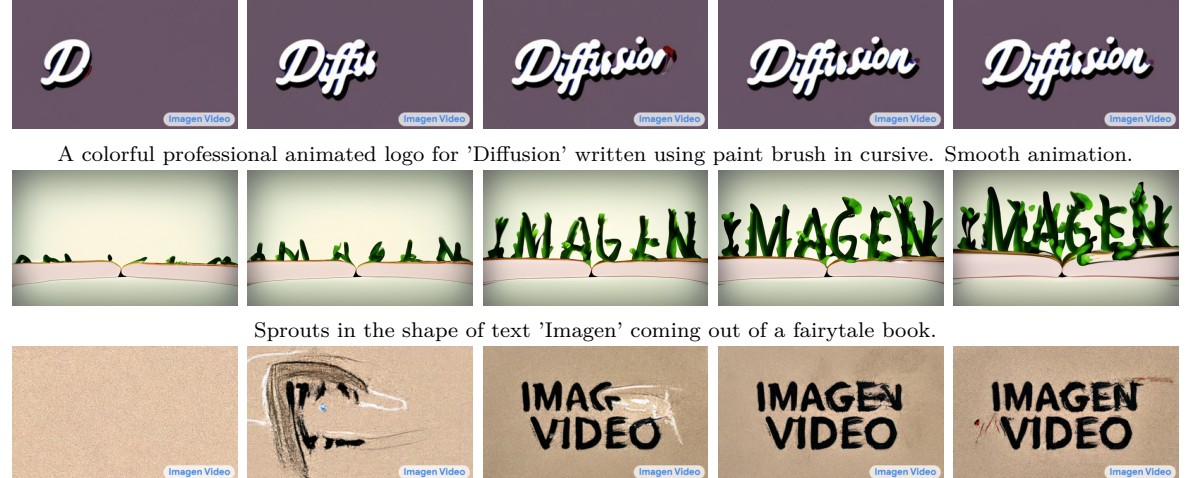

A colorful professional animated logo for 'Diffusion' written using paint brush in cursive. Smooth animation.

Sprouts in the shape of text 'Imagen' coming out of a fairytale book.

Thousands of fast brush strokes slowly forming the text 'Imagen Video' on a light beige canvas. Smooth animation.

Figure 10: Snapshots of frames from videos generated by Imagen Video demonstrating the ability of the model to render a variety of text with different style and dynamics.

exact, we believe Imagen Video shows that video models can serve as effective priors for methods that do force 3D consistency. Fig. 10 shows that Imagen Video is also reliably capable of generating text in a wide variety of animation styles, some of which would be difficult to animate using traditional tools. We see results such as these as an exciting indication of how general purpose generative models such as Imagen Video can significantly decrease the difficulty of high quality content generation.

## 3.2 Scaling

In Figure 11 we show that our base video model strongly benefits from scaling up the parameter count of the video U-Net. We performed this scaling by increasing the base channel count and depth of the network. This result is contrary to the text-to-image U-Net scaling results by Saharia et al. (2022b), which found limited benefit from diffusion model scaling when measured by image-text sample quality scores. We conclude that video modeling is a harder task for which performance is not yet saturated at current model sizes, implying future benefits to further model scaling for video generation.

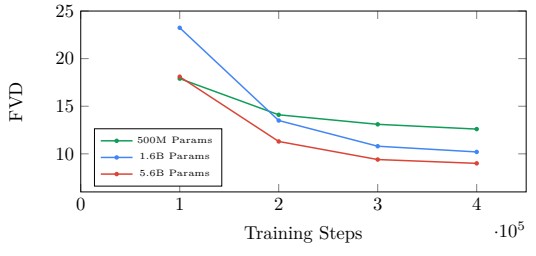

(a) Scaling Comparison on FVD scores.

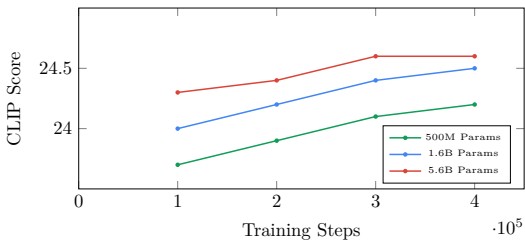

(b) Scaling Comparison on CLIP scores.

Figure 11: Scaling Comparison for the base 16×40×24 video model on FVD and CLIP scores (on 0-100 scale). Both FVD and CLIP scores are computed on 4096 video samples. We see clear signs of improvement on both metrics when scaling from 500M to 1.6B to 5.6B parameters.

## 3.3 Comparing prediction parameterizations

In early experiments we found that training $\epsilon$-prediction models (Ho et al., 2020) performed worse than $\mathbf{v}$-prediction (Salimans & Ho, 2022) especially at high resolutions. Specifically, for high resolution SSR models, we observed that $\epsilon$-prediction converges relatively slowly in terms of sample quality metrics and suffers from color shift and color inconsistency across frames in the generated videos. Fig. 12 shows the comparison between $\epsilon$-prediction and $\mathbf{v}$-prediction on a $80{\times}48 \rightarrow 320{\times}192$ video spatial super-resolution task. It is clear that $\epsilon$-parameterization produces worse generations than $\mathbf{v}$-parameterization. Fig. 13 shows the quantitative comparison between the two parameterizations as a function of training steps. We observe that $\mathbf{v}$ parameterization converges much more faster than $\epsilon$ parameterization.

## 3.4 Perceptual quality and distillation

In Table 1 we report perceptual quality metrics (CLIP score and CLIP R-Precision) for our model samples, as well as for their distilled version. Samples are generated and evaluated at 192×320 resolution for 128 frames at 24 frames per second. For CLIP score, we take the average score over all frames. For CLIP R-Precision (Park et al., 2021) we compute the top-1 accuracy (i.e. $R = 1$), treating the frames of a video sample as images sharing the same text label (the prompt). We repeat these over four different runs and report the mean and standard error.

We find that distillation provides a very favorable trade-off between sampling time and perceptual quality: the distilled cascade is about 18× faster, while producing videos of similar quality to the samples from the original models. In terms of FLOPs, the distilled models are about 36× more efficient: The original cascade evaluates each model twice (in parallel) to apply classifier-free guidance, while our distilled models do not, since they distilled the effect of guidance into a single model. We provide samples from our original and distilled cascade in Figure 14 for illustration.

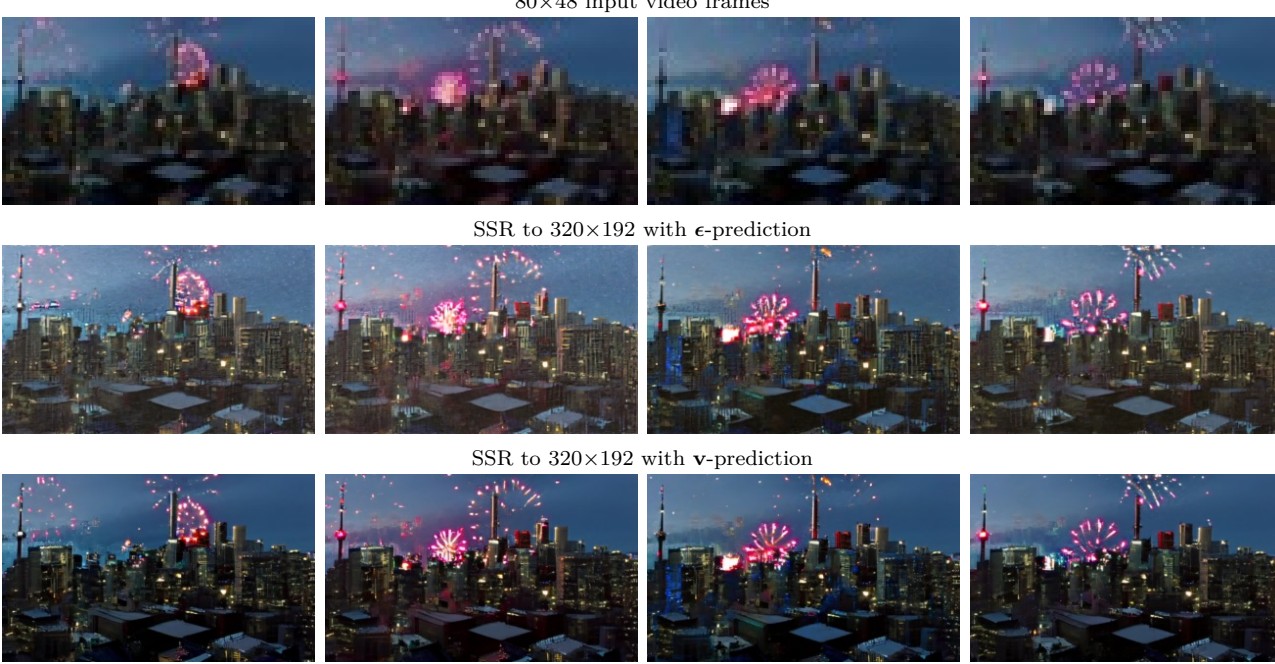

Figure 12: Comparison between $\boldsymbol{\epsilon}$-prediction (middle row) and $\mathbf{v}$-prediction (bottom row) for a $8{\times}80{\times}48{\rightarrow}8{\times}320{\times}192$ spatial super-resolution architecture at 200k training steps. The frames from the $\boldsymbol{\epsilon}$-prediction model are generally worse, suffering from unnatural global color shifts across frames. The frames from the $\mathbf{v}$-prediction model do not and are more consistent.

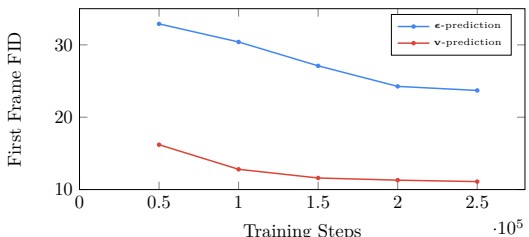

Figure 13: Comparison between $80{\times}48 \rightarrow 320{\times}192$ SSR models trained with $\boldsymbol{\epsilon}$- and $\mathbf{v}$-prediction parameterizations. We report FID evaluated on the first upsampled frame; FVD score is excessively noisy for the $\boldsymbol{\epsilon}$-prediction model. We observe that the sample quality of the $\boldsymbol{\epsilon}$-prediction model converges much more slowly than that of the $\mathbf{v}$-prediction model.

## 4 Limitations and Societal Impact

Generative modeling has made tremendous progress, especially in recent text-to-image models (Saharia et al., 2022b; Ramesh et al., 2022; Rombach et al., 2022). Imagen Video is another step forward in generative modelling capabilities, advancing text-to-video AI systems. Video generative models can be used to positively impact society, for example by amplifying and augmenting human creativity. However, these generative models may also be misused, for example to generate fake, hateful, explicit or harmful content. We have taken multiple steps to minimize these concerns, for example in internal trials, we apply input text prompt filtering, and output video content filtering. However, there are several important safety and ethical challenges remaining. Imagen Video and its frozen T5-XXL text encoder were trained on problematic data (Bordia & Bowman, 2017; Birhane et al., 2021; Bender et al., 2021). While our internal testing suggests much of explicit and violent content can be filtered out, there still exists social biases and stereotypes which are challenging

| Guidance $w$ | Base Steps | SR Steps | CLIP Score | CLIP R-Precision | Sampling Time |
|---|---|---|---|---|---|
| constant=6 | 256 | 128 | 25.19±.03 | 92.12±.53 | 618 sec |
| oscillate(15,1) | 256 | 128 | 25.02±.08 | 89.91±.96 | 618 sec |
| constant=6 | 256 | 8 | 25.29±.05 | 90.88±.50 | 135 sec |
| oscillate(15,1) | 256 | 8 | 25.15±.09 | 88.78±.69 | 135 sec |
| constant=6 | 8 | 8 | 25.03±.05 | 89.68±.38 | 35 sec |
| oscillate(15,1) | 8 | 8 | 25.12±.07 | 90.97±.46 | 35 sec |
| ground truth | | | 24.27 | 86.18 | |

Table 1: CLIP scores and CLIP R-Precision (Park et al., 2021) values for generated samples and ground truth videos on prompts from our test set. Cells highlighted in green represent distilled models. We compare three different combinations: original pipeline, distilled SR models on top of original base model, and fully distilled pipeline. The original base models use 256 sampling steps, and original SR models use 128 steps. All distilled models use 8 sampling steps per stage. Sampling from the original pipeline takes 618 seconds for one batch of samples, while sampling from the distilled pipeline takes 35 seconds, making the distilled pipeline about $18\times$ faster. We also explored two different classifier-free guidance settings for the base models: constant guidance with $w = 6$ and *oscillating* guidance which alternates between $w = 15$ and $w = 1$, following Saharia et al. (2022b). When using oscillating guidance, the fully distilled pipeline performs the same as the original model, or even slightly better. When using fixed guidance, our fully distilled pipeline scores slightly lower than the original model, though the difference is minor. Combining the original base model with fixed guidance and distilled super-resolution models produced the highest CLIP score. For all models, generated samples obtain better perceptual quality metrics than the original ground truth data: By using classifier-free guidance our models sample from a distribution tilted towards these quality metrics, rather than from an accurate approximation of the original data distribution.

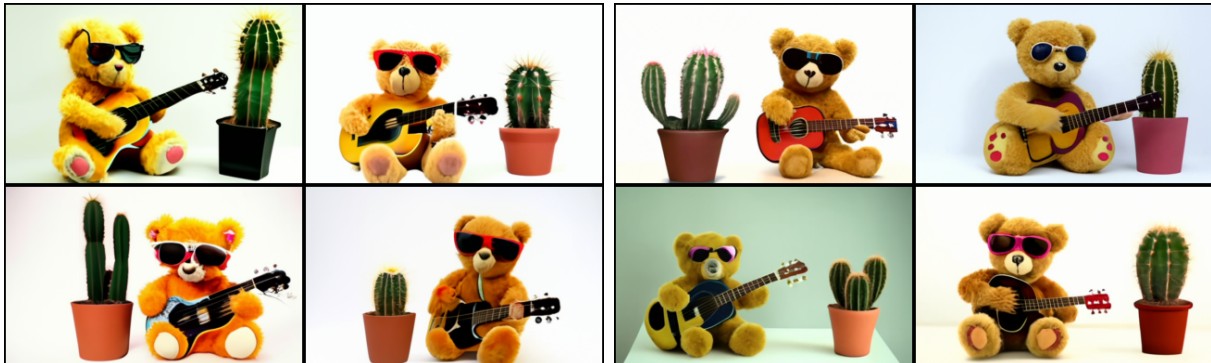

Figure 14: Frames from videos generated by Imagen Video for the text prompt "*A teddy bear wearing sunglasses playing guitar next to a cactus.*" The samples on the left are produced by our original model cascade, while the samples on the right are from our distilled cascade with 8 sampling steps per stage. Both used constant guidance with $w = 6$ and static clipping.

to detect and filter. We have decided not to release the Imagen Video model or its source code until these concerns are mitigated.

## 5 Conclusion

We presented *Imagen Video*: a text-conditional video generation system based on a cascade of video diffusion models. By extending the text-to-image diffusion models of *Imagen* (Saharia et al., 2022b) to the time domain, and training jointly on video and images, we obtained a model capable of generating high fidelity videos with good temporal consistency while maintaining the strong features of the original image system,

such as the ability to accurately spell text. We transferred multiple methods from the image domain to video, such as **v**-parameterization (Salimans & Ho, 2022), conditioning augmentation (Ho et al., 2022a), and classifier-free guidance (Ho & Salimans, 2021), and found that these are also useful in the video setting. Video modeling is computationally demanding, and we found that progressive distillation (Salimans & Ho, 2022; Meng et al., 2022) is a valuable technique for speeding up video diffusion models at sampling time. Given the tremendous recent progress in generative modeling, we believe there is ample scope for further improvements in video generation capabilities in future work.

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
