# OpenReview forum: "Imagen Video: High Definition Video Generation with Diffusion Models"
_TMLR — Rejected by TMLR_

### Review · Reviewer_v8qS · 2023-01-12

**Summary Of Contributions:**

This work proposed a text-conditional video generation model, named Imagen Video, based on a cascade of video diffusion models. The main idea is to extend the existing text-to-image diffusion model, named Imagen (Saharia et al., 2022), to the video domain, consisting of a frozen T5 text encoder (Raffel et al., 2020), a base video diffusion model (Ho et al., 2022), and interleaved spatial and temporal super-resolution diffusion models. Empirically, it demonstrates the simplicity and effectiveness of cascaded video diffusion models for high-quality video generation. Also, this work confirms that new findings in the text-to-image diffusion models can transfer to video generation, including v-parameterization, noise conditioning augmentation, and classifier-free guidance.

**Audience:**

Yes

**Broader Impact Concerns:**

The potential applications and the negative impacts have been well discussed in the paper.

**Claims And Evidence:**

Yes

**Requested Changes:**

Please see the weaknesses and minor concerns in the above.

**Strengths And Weaknesses:**

Strengths:
- With a cascade model, Imagen Video can generate high-definition videos of 128 frames with a resolution of 1280x768 at 24 frames per second. Particularly compared with previous works, the generation in this work is performed in a higher spatial resolution and a more diverse & less restricted domain.
- Many useful tricks, such as v-parameterization, noise conditioning augmentation, classifier-free guidance, and progressive distillation, have been explored and shown to be useful in video generation. Although the use of these tricks seems straightforward and the observations look not quite significant, it still sheds good insights into the video generation community.
- it also shows some good properties of text-based controllability, including 3D object understanding, text animations, and controlling artistic styles.

Weaknesses:
- The comparison with existing video diffusion models, in particular recently proposed large-scale models, such as CogVideo (Hong et al., 2022), and Make-A-Video (Singer et al., 2022), is not quite clear and thorough. For example, in the related work, the discussion with Make-A-Video is only about “building on a pretrained text-to-image model”. How about the generation capabilities and more detailed technical differences between Make-A-Video and Image Video?
- I’m not quite sure if TMLR allows adding videos or their links to the paper submission. If so, it is really good to watch the generated videos, in addition to the current static per-frame images.
- It would be good to also discuss the failure cases of Imagen Video, given the results presented in the paper are mostly about generated images.
- Typo: “much more faster” in the last sentence of Sec 3.3.

---

> ### Author Response · Authors · 2023-02-06
> **Response**
>
> - On comparison with existing models: we will add this to the paper. Briefly, CogVideo uses discrete autoregressive transformers on learned discrete tokens like VQVAEs, whereas Image Video uses continuous diffusion models with no learned latents. Both systems generate videos by starting at low frame rates then upsampling to high frame rates. Make-A-Video uses diffusion models and space-time factorized architectures for them, but a different training and sampling pipeline that emphasizes image pretraining: starting with CLIP encoding of text, Make-A-Video trains a prior network that generates image embeddings given text, trained on image-text pairs only; then they use image-only data to train models that generate low resolution images given those embeddings and super-resolution models following the low resolution; then they insert temporal layers and then fine-tune on unpaired video data. Image Video uses a joint training setup in which all models are conditioned on text via T5 embeddings and trained with both text-image pairs and text-video pairs by randomly mixing the two sources at training time; we did not explore training on unpaired data.
> - We will provide sample videos in the supplementary material.
> - Failure cases of Imagen Video include the inability to generate long sequences of actions given by long text prompts, largely due to the limitations of the training data and the length limit of the generated videos, as well as difficulties in achieving samples of extreme photorealism and highly precise details in actions and movements of objects. We believe these issues can be resolved by further scaling of model sizes and data.

---

### Review · Reviewer_3E2W · 2023-01-13

**Summary Of Contributions:**

- A brief but insightful analysis on the effect of v-prediction compared to $\epsilon$-prediction
   - demonstrated effects on convergence rates with respect to qualitative metrics
   - showed a few qualitative examples with distinct differences
- Model size analysis suggesting that video modeling can further benefit from larger models
- Provides further evidence for the use of progressive distillation for reducing diffusion model sampling time


**Audience:**

Yes

**Broader Impact Concerns:**

Broader impact claims appear sufficient

**Claims And Evidence:**

No

**Requested Changes:**

Please see suggestions as described in the "weaknesses" category above. The most important suggestions are those pertaining to additional detail in the experimental section regarding the justification of oscillated guidance. As is, I think some of the claims/statements in this work require further evidence or justification.

**Strengths And Weaknesses:**

Strengths:

- Writing was easy to follow
- Demonstrates successful construction of a large-scale model
- Provides further evidence for the efficacy of several existing strategies in diffusion model training
- In general, the model design seems fairly principled and well motivated

Weaknesses:

- Parts of the writing are a bit short on details. Because of the format of the submission, it’s reasonable to be a bit more verbose.
- Section 2.4 defines the v in v prediction, but doesn’t really put into context what v is. Better clarity could easily be achieved by mentioning v-prediction as in contrast to epsilon prediction (as is done later in the paper), and including the full denoising formula with respect to v.
In Section 2.6.2, a little more can be said about dynamic thresholding. Perhaps mention a little about how the dynamic threshold $s$ is chosen.
- Metric choices FID, FVD, CLIP, and CLIP R-Precision all have their limitations and assumptions. It would be best to briefly discuss these so that the reader has a better understanding of what it means when an improvement is demonstrated for each.
- Scaling discussion could use more explaining. Specifically, the authors mention contrary results to those previously reported. What are some possible reasons for this change in conclusion?
- Phrasing in Table 1’s caption seems to suggest, I assume incorrectly, that oscillating guidance is proposed in Saharia et al 2022b
- 2.6.2 claims dynamic thresholding helped with saturation, but was not sufficient in initial experiments. In what sense was it insufficient? It would also be nice to include some qualitative examples of where dynamic thresholding fixes saturation.
- 2.6.2 includes the statement: "We hypothesize that a constant high guidance weight at the start of sampling helps break modes with heavy emphasis on text, while oscillating between high and low guidance weights helps maintain a strong text alignment (via high guidance sampling step) while limiting saturation artifacts (via low guidance sampling step). "
   - While it may not be possible to conclusively determine if this is true, are there any experiments that can be included to corroborate even parts of this statement? This is important because a lot of the new insight provided by this paper are related to justifying the use of oscillated guidance.

Minor notes:

- Legend text in graphs needs to be larger
- 3.3 much more faster -> faster/much faster

---

> ### Author Response · Authors · 2023-02-06
> **Response**
>
> - Shortness of writing: thank you for this feedback; we will expand the text where requested.
> - On the explanation of v-prediction: we will insert text that explains v-prediction and its justification in the context of other parameterizations, such as epsilon-prediction. The dynamic thresholding parameter was chosen by a hyperparameter search for sample quality.
> - Discussion of FID, FVD, CLIP, and CLIP R-Precision: we will add discussion of these metrics as requested – for example, in the context of this work, FID measures quality and diversity of individual frames taken as images, FVD measures quality and diversity of videos, CLIP measures image-text alignment of frames.
> - Scaling discussion: when we scaled our video models, we found that sample quality (as measured by FVD and CLIP, for example) benefited from larger models, in contrast to findings in existing literature that text-to-image models did not benefit from scaling. The explanation we pose for this phenomenon is that our networks at their current scale are still underparameterized for the video generative modeling task, which is much more difficult than the previously mentioned image generative modeling tasks, for which networks at their current scale have reached a sufficient number of parameters.
> - Dynamic thresholding was insufficient in the sense that it was difficult to find a threshold that both helped sufficiently with saturation issues while preserving the quality of samples without introducing artifacts.
> - On oscillating guidance: we will incorporate samples that show how it helps sample quality. Generally, high guidance weights emphasize conditioning information but can degrade sample quality when the weight is driven too high because the resulting sample is too out-of-distribution; the low guidance weight steps project the sample back to a manifold which is closer to the training distribution. Thus alternating between high and low guidance weights is similar to a predictor-corrector method in which the low weight step corresponds to the corrector step.

---

### Review · Reviewer_Z9AN · 2023-01-21

**Summary Of Contributions:**

The paper describes a diffusion-based method for generating videos from text. The model builds up upon a base model to create higher-resolution (spatial and temporal) videos than previous work by including spatial and temporal superresolution module in the framework. It combines working principles from different approaches in order to achieve the final result. The result has been mainly been evaluated by showing a lot of examples of image sequences.

**Audience:**

Yes

**Broader Impact Concerns:**

Sociatal impact has been discussed. This has also been given as a reason to not release the model and code.

**Claims And Evidence:**

No

**Requested Changes:**

It is hard to follow the current manuscript to get the actual contributions and advantages of the method. The paper reads more like a technical report, describing a specific framework. However, still not detailed enough to reproduce it (e.g., architectural design details of the spatial and temporal upsampling modules). Clear comparison to related work are missing. Further, some temporal results in form of a video are necessary.

**Strengths And Weaknesses:**

### Strengths
Able to train from image-text datasets which are easier available than video-text datasets. However, that has also been done already in other works (Singer et al., 2022 ).

It can generate videos of higher temporal and spatial resolution.


### Weaknesses
The main technical contribution becomes not very clear. Each module described includes a reference to another work.

What is the used base model? Is it Imagen (Saharia et al., 2022) as mentioned in the conclusion or Video diffusion models (Ho et al, 2022) as mentioned in the introduction? It would be also clearer to add the references, e.g., in Figure 6, if some modules are related work.

The paper does not compare to any other method, e.g., Video diffusion models (Ho et al, 2022).

The paper claims that the proposed included spatial superresolution model is better/does not suffer from the artifacts as other existing superresolution methods. However, no comparison is shown. Also no comparison to using existing frame interpolation methods is shown.

It is unusual to start the paper with 3 full pages of results. In order to actually assess the claimed improved quality, video results would be much better suited. Including with and without the additional modules for spatial and temporal superresolution to assess the advantage of the proposed additions.

---

> ### Author Response · Authors · 2023-02-06
> **Response**
>
> - Our main technical contribution is the demonstration of the effectiveness of a diffusion-based video generation system at large scale in terms of sample quality and ability to generate coherent videos across a wide variety of scenes driven by text prompts, including surprising capabilities such as animated text rendering.
> - The base model is a video diffusion model at low spatial and temporal resolution, jointly trained on images and videos. We will clarify the relationship of each of the models to prior work.
> - Regarding artifacts of super-resolution: in the paper we write “our SSR models do not suffer from obvious artifacts that would occur from naively running super-resolution on independent frames.” This statement refers to the particular setting of performing video super-resolution by running image super-resolution on each frame separately, which would lead to artifacts due to an independence assumption over time. Our setup, by contrast, predicts multiple output frames conditioned on multiple input frames, so no such independence assumption is made.
> - We will incorporate more detail and architecture hyperparameters for each model in our pipeline, and we will provide video results in supplementary material. We will also run existing video super-resolution methods on our low-resolution generated videos and compare the results to our super-resolution models.

---

### Decision · Action_Editors · 2023-02-24

**Recommendation:** Reject

**Comment:**

The visual results of the proposed approach look great. Although the approach is a combination of existing approaches, novelty factor is not a major concern for TMLR acceptance. In my opinion, such type of papers' acceptance criteria should be on the clarity of the explanations for the engineering choices of the system, with clear experimental results (and even better with ablation studies) to support the engineering decisions. I made my decision on this paper based on these principles.

All three reviewers have asked for more details of the method and to provide additional visual evidence for the claims, suggestions include:
- links to the generated video -- the authors have promised a fix to respond to the first point;
- clarifications on some of the claims made for the engineering choices (see reviews);
- comparisons to other existing models in text-to-video generation domain -- I think it may be difficult to generate more comparisons to some other existing models, so this point does not have an easy fix.

On the second point, I agree with the reviewers that clarifications on some of the claims as well as some engineering details are indeed needed. When I read the paper, I feel I don't have enough information to reproduce the results even if I have access to thousands of GPUs and a gigantic dataset. The authors addressed the reviewers' questions via replies, but the reviewers felt it is unclear how these replies will be incorporated in the revised paper. Personally I am also not sure to what extent the engineering details can be shared in this manuscript.

In summary, while the visual results for the proposed Images Video system are very appealing, and I'm sure the authors are proud of their engineering achievements, the paper cannot be accepted in its current form due to the clarity concerns described above. If there is "major revision" option at TMLR I would certainly recommend that. Authors are welcome to revise and re-submit to TMLR, but again, it's on the authors' discretion about what kind of revision they would like to make. My suggestion would be to try to make the paper self-contained if possible, by e.g., referring to the network architecture used in the diffusion model (even if the model is from previous work). These kind of descriptions can be put into appendix, which will be much appreciated by the readers of this paper.

**Audience:**

Audience: researchers working in the field of generative models (especially diffusion models), multimodal machine learning, and video generation.
They would be interested in reading the papers, especially on the engineering details which is missing to a large extent.

**Claims And Evidence:**

The paper proposes a diffusion-based method for generating videos from text.

It combines working principles from different existing approaches (T5 for the text model, video diffusion models with text conditioning, super-resolution, classifier-free guidance, progressive distillation), in order to produce the final model that can generate high-resolution videos.

Experimental results are presented both qualitatively (sequences of frames from the generated videos) and quantitatively (FVD, CLIP score, and results on scaling laws and computational time).

The image quality is great and convincing. However, reviewers also pointed out that, no comparison is made for other existing approaches.